# Slug Regime Transitions in a Two-Phase Flow in Horizontal Round Pipe. CFD Simulations

**Vitaly Sergeev, Nikolai Vatin** **, Evgeny Kotov, Darya Nemova * and Svyatoslav Khorobrov**

Institute of Civil Engineering, Peter the Great Saint Petersburg Polytechnic University,
195251 Saint Petersburg, Russia; vitaly.sergeev@spbstu.ru (V.S.); vatin_ni@spbstu.ru (N.V.);
kotov_ev@spbstu.ru (E.K.); horobrov_sv@spbstu.ru (S.K.)
* Correspondence: nemova_dv@spbstu.ru; Tel.: +7-921-890-02-67

**Abstract:** The main objective of the study is to propose a technical solution integrated into the pipeline for the transition of the flow regime from slug to bubbly two-phase flow. The object of research is isothermal two-phase gas–Newtonian-liquid flow in a horizontal circular pipeline. There is local resistance in the pipe in the form of a streamlined transverse mesh partition. The mesh partition ensures the transition of the flow from the slug regime to the bubbly regime. The purpose of the study is to propose a technical solution integrated into the pipeline for changing the flow regime of a two-phase flow from slug to bubbly flow. The method of research is a simulation using computational fluid dynamics (CFD) numerical simulation. The Navier–Stokes equations averaged by Reynolds describes the fluid motion. The $k$-$\varepsilon$ models were used to close the Reynolds-averaged Navier–Stokes (RANS) equations. The computing cluster «Polytechnic—RSK Tornado» was used to solve the tasks. The results of simulation show that pressure drop on the grid did not exceed 10% of the pressure drop along the length of the pipeline. The mesh partition transits the flow regime from slug to layered one, which will help to increase the service life and operational safety of a real pipeline at insignificant energy costs to overcome the additional resistance integrated into the pipeline.

**Keywords:** two-phase flow; two-phase flow regimes; regime transitions; pipe flow; oil pipeline; plug flow; slug flow; intermittent flow; computational fluid dynamics

## 1. Introduction

### 1.1. Two-Phase Gas–Liquid Flow in Horizontal Round Pipe Section

The two-phase gas–liquid flow with the slug flow pattern in round macro- and microchannel or pipes is frequently detected in many technical systems; for example, in the oil and gas industry, in power plants, sewage systems, and metallurgy [1–5].

Two-phase flows and mass porosity are intensively studied for the cases of flow in macro- and microchannels. A detailed review of microchannel flows is given in [6]. Flow in microchannels was not considered in this article.

The simplest cases of two-phase flows in a round pipe are vertical upward two-phase flow and vertical downward two-phase flow. A feature of two-phase gas–liquid flow in inclined pipes [7,8] is the position of the gas and liquid phases of the flow asymmetric relative to the pipe axis as a result of the effect of gravity on the flow. In this case, the upward and downward flows in the inclined pipe have different characteristics. A feature of horizontal two-phase flow is also an asymmetrical distribution of two phases.

Horizontal two-phase gas–liquid flow modes are mainly classified as [1,9]:

- Stratified flow: In this mode, the phases are separated along the channel height. Layered flow is observed at low velocities of gas and liquid.
- Wavy flow: As the velocity of the gas phase of the flow increases, waves are formed at the gas–liquid interface. Waves propagate in the direction of the flow.
- Bubbly flow: The gas phase is distributed in the form of individual bubbles in a liquid medium. Most often, bubbles are mixed with liquid at the top of the pipe.
- Plug flow: This flow is an intermittent flow, observed at low flow rates of gas and liquid. The mode is characterized by elongated gas bubbles (plugs).
- Slug flow: This flow is an intermittent flow. The flow is characterized by chaotic formation and movement of gas bubbles and a less distinct gas–liquid interface. Plug and slug flow patterns are often classified as intermittent flow.
- Churn flow: Occurs when the flow rate increases. The bubble shells break, and an emulsion appears. The flow regime is transitional between the slug and annular.
- Annular flow: Higher gas velocities create liquid films on the channel walls.

The visual patterns of horizontal gas–liquid flow modes are presented in [10]. The velocity difference between the two phases increases at a higher gas quantity [11]. The flow pattern in corrugated pipes is significantly more complex than flow in smooth ones [12]. In this paper, only the flow in smooth pipes is considered.

### 1.2. Experimental Investigation of Slug Regime of Gas–Liquid Flow in Horizontal Round Pipe Section

The slug flow mode can cause dangerous vibrations of the pipe and associated process equipment, large pressure surges, and, as a consequence, rapid mechanical wear of pipeline elements [13]. That is why it is necessary to clearly understand which flow regime of the two-phase mixture will be implemented with specific parameters in the inlet section, and what measures must be taken to avoid the occurrence of undesirable flow regimes.

There are two variants of the slug flow regime, called circulation and bypass [1]. These regimes differ in streamlines drawn around the slug (Figure 1).

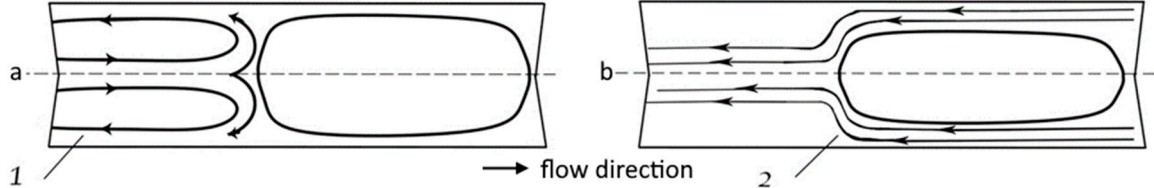

**Figure 1.** Two variants of the slug flow regime, called circulation and bypass: a—circulating flow; b—bypass flow; streamlines of the liquid: 1—in circulation; 2—in bypass modes.

For technical applications of macro-tube flows, it is essential to know the pressure drop [14], the presence or absence of a slug flow pattern, and the frequency of the slugs. This problem has not been adequately solved for straight pipes [10,15,16], although there is research on the more complex case of air–water two-phase flows in horizontal U-bend pipes [17–20]. The slug regime in a two-phase gas–liquid flow in horizontal round pipes was experimentally investigated with different methods [10,21–24].

The transition from plug to slug flow in a horizontal pipe was experimentally studied with electrical capacitance tomography [21]. The researchers used air and silicone oil for two-phases flow simulation. The horizontal gas–oil flow was studied with non-intrusive electrical capacitance tomography measurements [10]. The frequency and time span of slugs was assessed. The detailed flow visualization studies of air–water two-phase flow in a round pipe was performed with a high-speed video camera in a wide range of two-phase flow conditions [22–24].

Ultrasonic detection was applied for moving slug interfaces detection in gas–liquid two-phase flow [25].

Two-phase flow regime transition mechanisms were exploited with high-resolution virtual experiments [26]. It was shown that the transition out of the slug flow regime could take at least two pathways:

- Interfacial wave-induced instability development in the Taylor bubble, leading to its disintegration.
- Intense bubble shearing at the tail of the bubble.

The estimation of pressure drop in two-phase gas–non-Newtonian liquid flow in long horizontal pipes was made with the multiple-layer perceptron artificial neural network model [27]. The model is based on a large number of experimental data. The input parameters of the model are a wide range of operating conditions, pipe diameters, and fluid characteristics. The authors claim that the estimations are in good agreement with the experimental results; but, such a model is a way of interpreting previously obtained experimental data and does not have predictive properties. In the absence of experimental data on two-phase flow in a pipe with local resistance, this kind of model could not be used for prediction of flow regime's transitions on local hydraulic resistance element.

The statistical approach for the slug flow often aims to predict the slug frequency [14]. Statistical features of the flow in horizontal liquid–gas slug flow was experimentally investigated [28]. The probability density functions of the slug frequency and the translational velocity was determined. Based on a detailed review of slug frequency for a gas–liquid plug flow, a new empirical correlation has been proposed [16].

Despite the large amount of experimental data obtained by researchers, there is no theory and method for calculating such flows. The methods for calculating the pressure drop and flow regime transition of a multiphase flow are mainly empirical and semi-empirical. The application of these methods is mainly limited.

Two-phase flow is simulated using the computational fluid dynamics (CFD) method [29–32] and others. Due to the development of methods for numerical simulation of multiphase flows, it is possible to achieve high accuracy of simulation results. So, for example, using the Reynolds-averaged Navier–Stokes (RANS) equations approach and the volume of fluid (VOF) method, it is possible to reproduce any flow regime of a multiphase flow by varying the boundary conditions at the pipeline inlet. This approach makes it possible to eliminate flow regimes unfavorable for technological equipment.

However, we did not find, in the available publications, work on numerical simulation of two-phase flow with a local hydraulic resistance element that changes the flow regime from slug to bubbly.

*1.3. The Objectives of the Study*

The main objective of the study is to propose a technical solution integrated into the pipeline for the flow regime's transition from slug to bubbly two-phase flow. The resulting solution will increase the service life and improve the reliability of process pipelines used in power plants in the oil and gas industry.

Research objectives:

- Reproduce the slug flow regime of a two-phase flow in a horizontal circular smooth pipeline using the CFD simulation method.
- Determine the location of the slug.
- Propose a technical solution that can be integrated into the pipeline to ensure a change in the flow regime.

The research object is isothermal two-phase gas–Newtonian-liquid flow in a horizontal circular smooth pipeline.

The research subjects are flow rate, pressure drop, phase composition of the flow, and flow regime.

## 2. Materials and Methods

The research method was the numerical simulation in the ANSYS software package.

*2.1. Description of the Design Model*

　　A three-dimensional two-phase flow in a straight horizontal pipe was considered. Two geometry options with identical boundary conditions were considered.

2.1.1. Geometry and Mesh Partition

　　The design area was a straight pipe with a diameter of 0.08 m and a length of 8 m in two versions:

- In the first version of the geometry, there were no local modifications inside in the pipe (Figure 2).
- In the second version of the geometry, the mesh partition was located at a distance of 3.995 m (Figure 3). The distance between the partitions was 0.005 m, their thickness was 0.002 m, and their number was 11. The mesh partition is shown in Figure 4. The length of the pipe fragment with mesh partitions was 0.01 m.

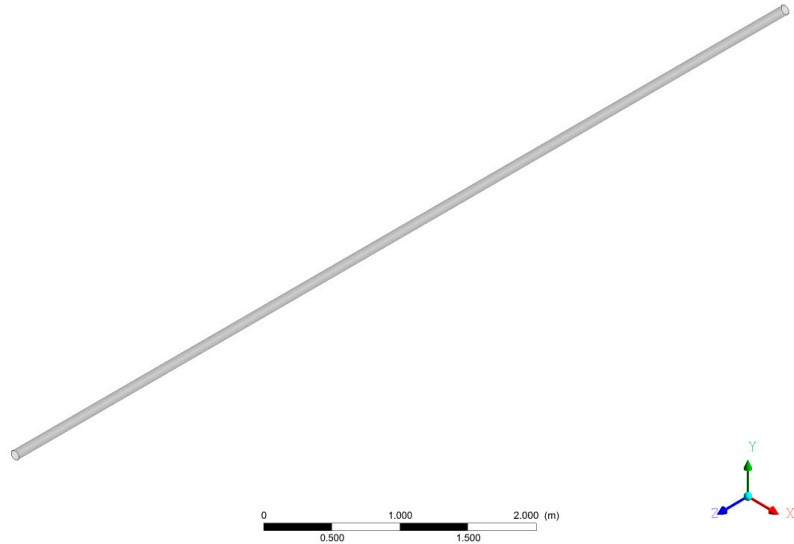

**Figure 2.** Geometry of the computational domain: the first version.

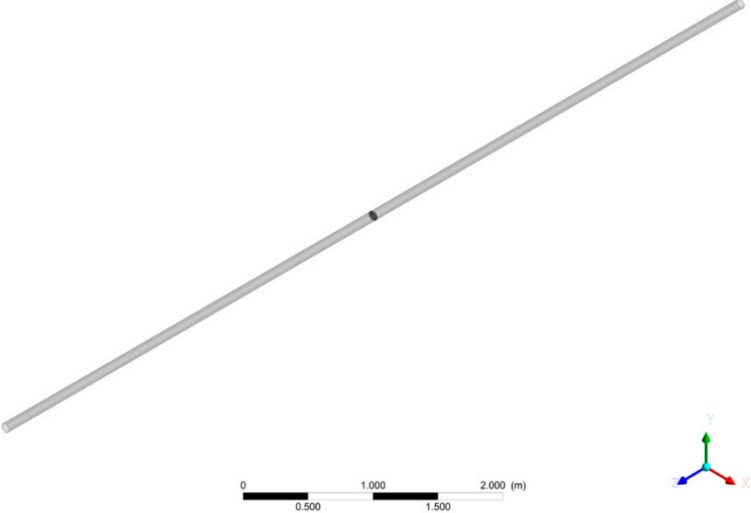

**Figure 3.** Geometry of the computational domain: the second version.

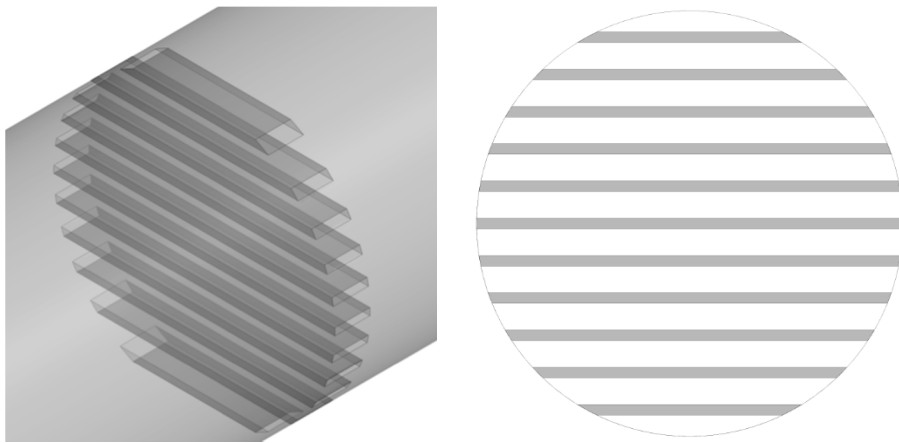

**Figure 4.** Mesh partition construction.

For each of the geometry options, a computational grid was built, providing solution grid convergence. For option 1, the grid was block-structured; for option 2, it was combined: unstructured in the vicinity of the grid, structured in the rest of the area (Figures 5 and 6).

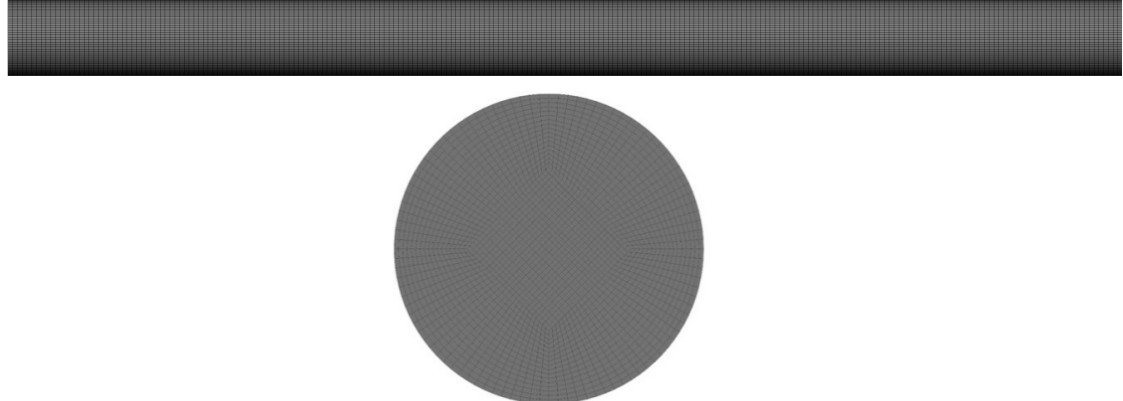

**Figure 5.** Block-structured grid for first version.

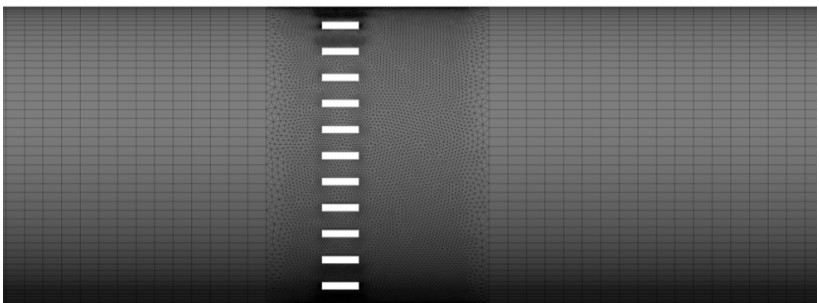

**Figure 6.** Hybrid mesh for second version.

The following parameters can characterize every computational grid:

(1)  Size (number of nodes and number of elements).
(2)  More elements usually improve the accuracy of the solution, but up to a certain limit.
(3)  Maximum and minimum angle at the cell face. The more these angles deviate from 90 degrees, the more skewed the cells will be, and the less accurate the uncorrected solution will be.
(4)  The aspect ratio of the grid cell and the ratio of the cell volumes. In general, if these parameters are too large, the solution accuracy decreases, due to the large non-uniformity of the flow parameters.

(5)   Compactness of the mesh. The smaller this parameter, the more detailed the mesh. The larger this parameter, the more elements correspond to one node.

The article does not aim to find the computational grid size that would ensure grid size independence. The computational grid satisfied the following criteria. Suppose the refinement of the grids by half in all three directions does not change the results by more than 1% relative to the previous results. In that case, the computational grid is considered acceptable, and provides a grid size-independent solution. The characteristics of the computational grids for types 1 and 2 are shown in Table 1.

**Table 1.** Parameters of computational grids.

| Parameters | Type 1 | Type 2 |
| --- | --- | --- |
| Number of nodes | 3,803,976 | 4,165,141 |
| Number of elements | 3,720,000 | 5,823,880 |
| Maximum angle at cell face | 90 | 50.04 |
| Minimum angle at cell face | 49.68 | 16.03 |
| Cell aspect ratio | 4.52 | 4.52 |
| Cell volume ratio | 1.37 | 178.75 |
| Compactness of mesh | 8 | 42 |

### 2.1.2. Environment Setting

The simulated medium in the computational domain for both options was two-phase. The main phase was air with a density of 1.225 kg/m$^3$ and a viscosity of 0.00017894 Pa·s; the additional medium was water with a density of 998.1 kg/m$^3$ and a viscosity of 0.001003 Pa·s (Table 2).

**Table 2.** Physical properties of the simulated phases.

| | Gas | Water |
| --- | --- | --- |
| Density, kg/m$^3$ | 1.225 | 998.1 |
| Dynamic viscosity, Pa·s | 0.00017894 | 0.001003 |

The interaction of media was described by the volume of fluid model, taking into account the surface tension between the phases.

### 2.1.3. Boundary Conditions

The following boundary conditions were used to solve the problem of flow of a multiphase medium: an inlet boundary condition with a profile of the velocity and concentration of phases, an outlet boundary condition, and a sticking condition. In both considered cases, the boundary conditions were identical.

A profile of the velocity and concentration of phases was established at the inlet to the computational domain. The setting was in agreement with the initial distribution of these two parameters. The profile was set using the user-defined function. There was air in the upper half of the pipe relative to the OY axis, and water in the lower half (Figure 7). Air velocity was 3.26 m/s, and water velocity was 0.4 m/s. At the initial moment, the phase velocities and their concentrations were the same throughout the pipe.

At the outlet from the computational domain, the condition of zero overpressure was set.

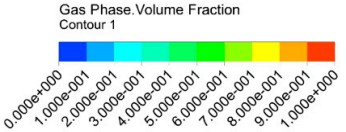

**Figure 7.** Initial distribution of phase concentrations.

### 2.1.4. Applied Turbulent Flow Models

Simulation of fluid flow was based on the calculation of the Navier–Stokes equations. The Navier–Stokes equations are a set of partial differential equations which describe the motion of viscous fluid. Below are the equations to be solved for the fluid motion model.

Continuity equation:

$$\frac{\partial \rho}{\partial t} + \vec{\nabla} \cdot (\rho V) = 0 \tag{1}$$

Momentum equation (according to Navier–Stokes equation):

$$\frac{\partial (\rho V)}{\partial t} + \vec{\nabla} \cdot (\rho V \times V) = \vec{\nabla} \cdot \left(-P\hat{I} + \hat{\tau}_{eff}\right) + \rho F \tag{2}$$

where:

$$\hat{\tau}_{eff} = \mu \left(2\hat{e} - \frac{2}{3} \left(\vec{\nabla} \cdot V\right)\hat{I}\right) \tag{3}$$

$$e_{ij} = \frac{1}{2} \left(\nabla_j V_i + \nabla_i V_j\right) \tag{4}$$

Energy equation:

$$\frac{\partial (\rho h)}{\partial t} + \vec{\nabla} \cdot (\rho V h) = \vec{\nabla} \cdot J_{q,eff} + \frac{dP}{dt} + \hat{\tau} : \hat{e} + \rho \varepsilon \tag{5}$$

$$J_{q,eff} = -\frac{\lambda}{C_p} \vec{\nabla} h \tag{6}$$

Since the velocities of water and air were high enough, the flow was assumed to be turbulent. Fluid motion was described by the Reynolds-averaged Navier–Stokes (RANS) equations [33].

The $k$-$\varepsilon$ model with scalable wall functions [34] was used to close the RANS equations calculation scheme. The boundary conditions for the turbulence model were standard: 5% turbulence intensity, and the ratio of turbulent viscosity to molecular viscosity equal to 10 at the inlet and outlet boundaries. The standard constants were used in the $k$-$\varepsilon$ model that Fluent offers by default.

Since there was a need to monitor the interface to determine the multiphase flow regime accurately, it was decided to use the volume of fluid (VOF) model. This model supplements the Navier–Stokes equations with the concentration equation for each of the phases in the cell of the computational grid. Thus, in each of the cells, there was one of either the phase or an interface. The VOF method was not very demanding on resources, and was one of the frequently used methods for this class of tasks [35]. For more accurate tracking of the behavior of the interface, the VOF method was improved, taking into account the surface tension between the phases. Besides, the calculation of the flow was carried out under the action of a gravitational force along the Y-axis.

Since the interface changes dynamically, the problem was unsteady in the time domain. The used time step range was from 10 ms to 1 ms. The time step was adaptive and generally close to the minimum value of 1 ms.

The time step was adjusted based on the condition that the solution was independent of the time step size.

## 2.2. Calculation Method

### 2.2.1. Numerical Methods

The ANSYS Fluent method for solving convective–diffusion equations is based on the finite volume method. The finite volume method assumes the integration of the equations of fluid motion

and the transfer of scalar quantities over the volumes of the cells of the computational grid [36]. By Gauss's theorem, for an arbitrary vector or tensor quantity *F*,

$$\int_{\Omega\text{cell}} \left(\vec{\nabla}\cdot F\right) d\Omega = \sum_{i=faces} (F_i\cdot k_i)\cdot\Delta S_i \tag{7}$$

Here, $k_i$ is the normal vector. The integration of the equations to be solved in the cell is performed by summing the fluxes of mass, momentum, energy, and other quantities calculated on the edges of the cells. Since each face separates two adjacent cells, the corresponding flow enters the discrete equations for both cells. This ensures the conservation rules of the sought values in the computational domain.

The possibility of fluid motion equations' integration at a one-time step could be demonstrated as follows. Consider the Euler equations (we neglect the viscous terms). The continuity equation and the momentum equation were integrated over the computational cell which is the polyhedron:

$$\frac{\rho^{n+1}-\rho^n}{\tau} + \frac{1}{\Omega}\sum_{faces}\left(\rho^{n+1}V^{n+1}\cdot k\cdot\Delta S\right) = 0 \tag{8}$$

$$\frac{\rho^{n+1}V^{n+1}-\rho^n V^n}{\tau} + \frac{1}{\Omega}\sum_{faces}(\rho^n V^n\cdot k\cdot\Delta S)V^{n+1} = -\vec{\nabla}P^{n+1} + \rho^n F^n \tag{9}$$

Equation (9) was rewritten as follows:

$$\frac{\rho^{n+1}V^{n+1}-\widetilde{\rho}^{n+1}\widetilde{V}^{n+1}}{\tau} + \frac{\widetilde{\rho}^{n+1}\widetilde{V}^{n+1}-\rho^n V^n}{\tau} + \frac{1}{\Omega}\sum_{faces}(\rho^n V^n\cdot k\cdot\Delta S)\widetilde{V}^{n+1}$$
$$= -\vec{\nabla}P^{n+1} + \vec{\nabla}P^n - \vec{\nabla}P^n + \rho^n F^n \tag{10}$$

and divided (split) into two equations:

$$\frac{\widetilde{\rho}^{n+1}\widetilde{V}^{n+1}-\rho^n V^n}{\tau}\frac{1}{\Omega}\sum_{faces}(\rho^n V^n\cdot k\cdot\Delta S)\widetilde{V}^{n+1} = -\vec{\nabla}P^n + \rho^n F^n \tag{11}$$

$$\frac{\rho^{n+1}V^{n+1}-\widetilde{\rho}^{n+1}\widetilde{V}^{n+1}}{\tau} = -\vec{\nabla}P^{n+1} + \vec{\nabla}P^n \tag{12}$$

The «tilde» sign denotes intermediate solutions. The mass velocity at $n + 1$ time intervals are expressed from Equation (12) as:

$$\rho^{n+1}V^{n+1} = \widetilde{\rho}^{n+1}\widetilde{V}^{n+1} + \tau\left(-\vec{\nabla}P^{n+1} + \vec{\nabla}P^n\right) \tag{13}$$

The Poisson equation for the pressure obtained by substituting Equation (13) into the discrete continuity Equation (1):

$$\frac{\rho^{n+1}-\rho^n}{\tau} + \frac{1}{\Omega}\sum_{faces}\left(\rho^{n+1}\widetilde{V}^{n+1}\cdot k\cdot\Delta S\right) = \tau\left(\Delta P^{n+1} + \Delta P^n\right) \tag{14}$$

An iterative scheme for simultaneous solution of the Navier–Stokes, energy, concentration, and free surface equations could be based on an implicit splitting algorithm in physical variables. The scheme could be explained in eight following steps.

Step 1:

The fulfilment of the discrete continuity equation was checked:

$$\frac{\rho^n - \rho^{n-1}}{\tau} + \frac{1}{\Omega} \sum_{faces} \left( \rho^n V^n \cdot k \cdot \Delta S \right) = 0 \tag{15}$$

Here the non-stationary term was estimated as follows:

$$\frac{\partial \rho}{\partial t} = \frac{\rho^n - \rho^{n-1}}{\tau} = \frac{\rho(T^n) - \rho(T^{n-1})}{\tau} \tag{16}$$

Step 2:

If Equation (15) was not fulfilled at least in a small part of the computational domain, then the equation for the pressure was solved:

$$\frac{\rho^n - \rho^{n-1}}{\tau} + \nabla(\rho^n V^n) = \tau\left( \Delta \widetilde{P}^{n+1} - \Delta P^n \right) \tag{17}$$

Step 3:

After finding the intermediate pressure field $\widetilde{P}^{n+1}$ the "conservative" (satisfying the continuity equation), the intermediate velocity field was calculated $\widetilde{V}^{n+1}$:

$$\rho^n \widetilde{V}^{n+1} = \rho^{n-1} V^n + \tau\left( -\vec{\nabla}\widetilde{P}^{n+1} + \vec{\nabla}P^n \right) \tag{18}$$

The intermediate pressure field $\widetilde{P}^{n+1}$ was not used anywhere in subsequent calculations.

Step 4:

The equation for the velocity was solved:

$$\begin{aligned}
&\frac{\rho^n \widetilde{\widetilde{V}}^{n+1} + \rho^{n-1} V^n}{\tau} + \frac{1}{\Omega} \sum_{faces} \left( \rho^n \widetilde{V}^{n+1} \cdot k \cdot \Delta S \right) \widetilde{\widetilde{V}}^{n+1} \\
&= -\vec{\nabla}P^n + \frac{1}{\Omega} \sum_{faces} \hat{\tau}_{eff} \cdot k \cdot \Delta S + \rho^n F^n
\end{aligned} \tag{19}$$

Here, the viscous stress tensor was determined by the intermediate velocity field $\widetilde{V}^{n+1}$. The second intermediate velocity fields $\widetilde{\widetilde{V}}^{n+1}$ were found as the result of solving Equation (19). In subsequent calculations, the field $\widetilde{V}^{n+1}$ was not used, so in the program the array $\widetilde{\widetilde{V}}^{n+1}$ was replaced by the array $\widetilde{V}^{n+1}$.

Step 5:

The equation for pressure was solved:

$$\frac{\rho^n - \rho^{n-1}}{\tau} + \vec{\nabla}\left( \rho^n \widetilde{\widetilde{V}}^{n+1} \right) = \tau\left( \Delta P^{n+1} - \Delta P^n \right) \tag{20}$$

Step 6:

$$\rho^n V^{n+1} = \rho^{n-1} V^n + \tau\left( -\vec{\nabla}P^{n+1} + \vec{\nabla}P^n \right) \tag{21}$$

Step 7:

The energy Equation (5) was solved, in which the field $V^{n+1}$ was used.

Step 8:

Density was determined as:

$$\rho^{n+1} = \rho\left( T^{n+1} \right) \tag{22}$$

The solution at the $n + 1$-st time level can be improved by repeating the described procedure. The second and subsequent passes will differ from the first in the following:

(a)  Lack of steps 1, 2, and 3.
(b)  Applied previous mass velocity $\rho^{n+1}V^{n+1}$ on the edges of the cells in the Equation (4).
(c)  Applied the previous density $\rho^{n+1}$ in non-stationary terms:

$$\frac{\partial \rho}{\partial t} = \frac{\rho^{n+1} - \rho^n}{\tau} \tag{23}$$

$$\frac{\partial (\rho V)}{\partial t} = \frac{\rho^{n+1}\widetilde{V}^{n+1} - \rho^n V^n}{\tau} \tag{24}$$

This numerical algorithm was highly robust. The results obtained are in good agreement with the experimental data. Stable grid convergence was observed.

### 2.2.2. Computational Scheme

The numerical simulation was performed using separate solving (i.e., the continuity equations and the equations of motion were solved separately) and using the pressure-implicit with splitting of operators (PISO) algorithm to connect both parts of the solver. For the equations of motion and the turbulence model, an upstream sampling scheme was used in the space of the second order of accuracy; the PRESTO! scheme [37] was used for the continuity equation; the Geo-Reconstruct scheme [38] was used for the equations of phase concentration in VOF model.

Time discretization of the equations was carried out with the first order of accuracy for higher algorithm convergence.

### 2.2.3. The Numerical Processing

The numerical simulation was made in the computing cluster «Polytechnic RSC Tornado», Peter the Great St. Petersburg Polytechnic University, St. Petersburg, Russian Federation. Each task was launched on four nodes, with direct liquid cooling of the Tornado series, each with two Intel Xeon E5-2697 v3 CPUs (14 cores, 2.6 GHz) and 64 GB of DDR4 RAM.

The average time to compute 0.005 s in real-time ranged from 35 to 60 min, depending on the computational grid used.

## 3. Results and Discussions

### 3.1. Pressure and Piezometric Lines

#### 3.1.1. The Pipe Without Mesh Partitions

At first, pressure and piezometric lines were calculated for the first pipeline option (without a mesh partition). Calculations were performed separately for the liquid and gas phases.

The main parameters of the liquid phase and the parameters of the design pipeline are presented in Section 2.1.2. Since the phase velocities in the inlet and outlet sections are the same, the pressure drop was equal to the length loss, and was determined by the formula:

$$h_l = \lambda \frac{LV^2}{D2g} \tag{25}$$

where:

$\lambda$ is the drag coefficient (take 0.039);
$L$ is pipeline length (m);

$D$ is pipeline diameter (m);

$V$ is the fluid velocity in the pipeline (m/s);

$g$ is gravity acceleration (m/s$^2$).

The pressure drop along the length for liquid and gas is:

$$h_{l(lq)} = \lambda \frac{LV^2}{D2g} = 0.039 \frac{8 \cdot 0.4^2}{0.08 \cdot 2 \cdot 9.81} = 0.03 \text{ m} \tag{26}$$

$$h_{l(g)} = \lambda \frac{LV^2}{D2g} = 0.039 \frac{8 \cdot 3.26^2}{0.08 \cdot 2 \cdot 9.81} = 2.11 \text{ m} \tag{27}$$

Based on the data obtained, pressure and piezometric lines were constructed for a pipeline without a mesh partition (Figure 8).

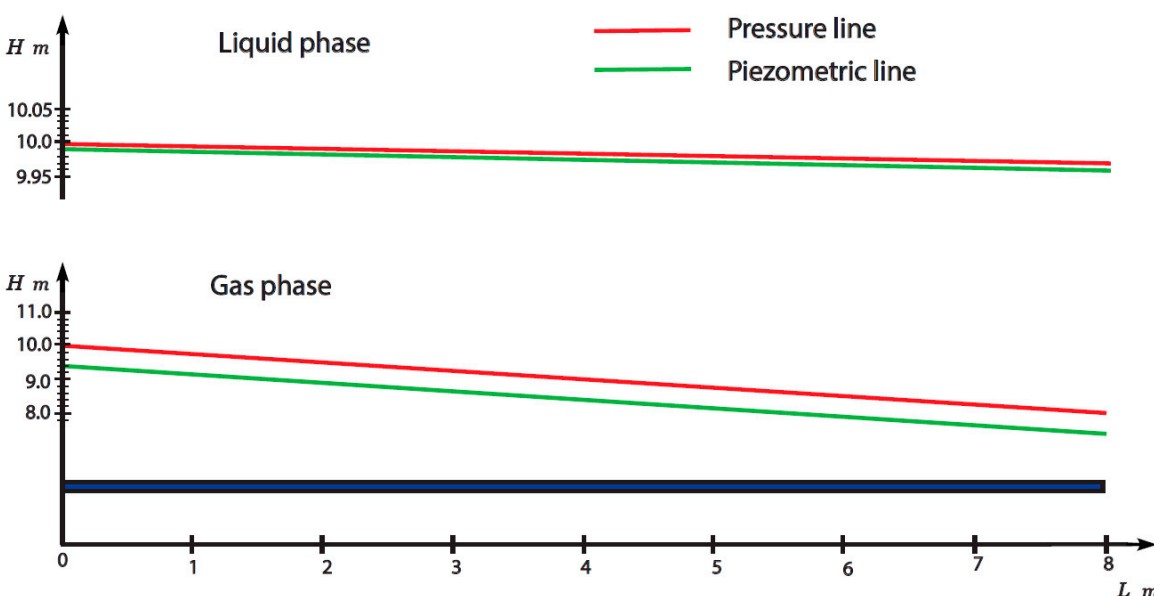

**Figure 8.** Pressure and piezometric lines for gas and liquid phases (pipeline without mesh partition).

### 3.1.2. Pipeline with Mesh Partition

In the second variant, a mesh partition was located in the middle of the pipeline. Its parameters are presented in Section 2.1.1. First, the pressure drop in the pipeline was calculated using the following formula:

$$\Delta h = h_l + h_m \tag{28}$$

$$h_{\mathrm{M}} = \zeta \frac{V^2}{2g} \tag{29}$$

where $\zeta$ is the local drag coefficient.

The pressure drops on the mesh partition for liquid and gas are described as:

$$h_{m(lq)} = 0.6 \cdot \frac{0.4^2}{2 \cdot 9.81} = 0.0048 \text{ m} \tag{30}$$

$$h_{m(g)} = 0.6 \cdot \frac{3.26^2}{2 \cdot 9.81} = 0.33 \text{ m} \tag{31}$$

The pressure drop along the length of the pipeline is the same as in the previous variant (without mesh partition).

Based on the data obtained, the piezometric and pressure lines were calculated (Figure 9) for variant 2 (the pipeline with the mesh partition).

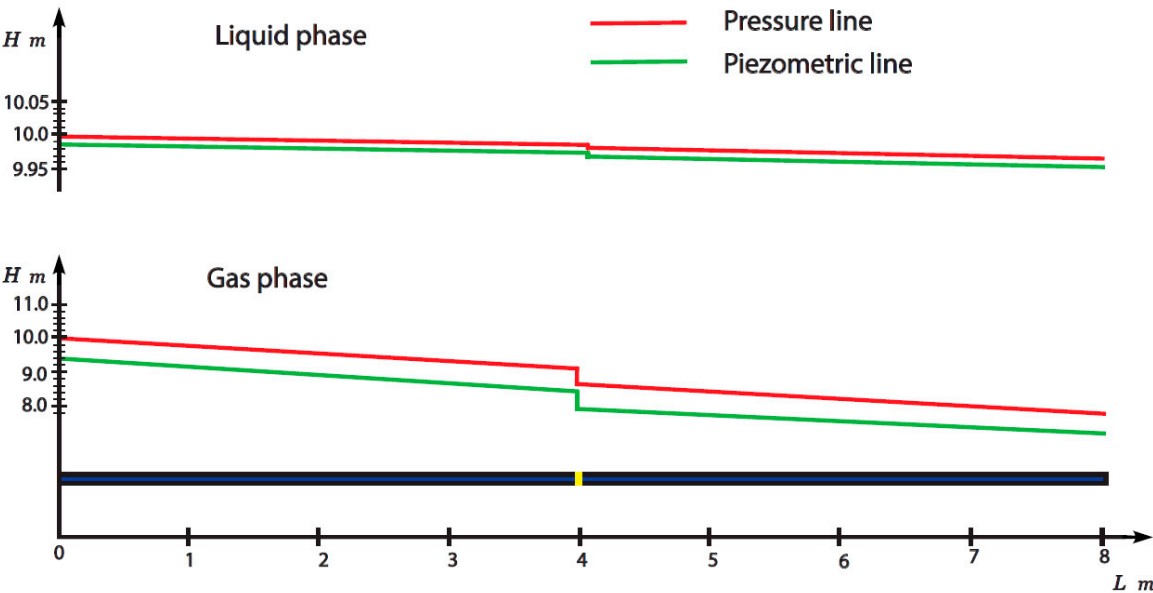

**Figure 9.** Pressure and piezometric lines for gas and liquid phases (the pipeline with the mesh partition).

### 3.2. Fields of Pressure, Velocity, and Concentration of Phases

Tables 3–8 show the evolution of pressure, velocity, and phase concentration fields for both considered cases. Consider the evolution of fields for a setting without a mesh partition. Prior to the formation of the slug regime, special attention has to be paid to the fields of velocity and concentration. From the concentration field at the initial moment of time, it can be seen that a layered flow regime was observed in the inlet section of the pipe, corresponding to the boundary condition set at the inlet. At the same time, it was seen that the speed of the light phase located in the upper part of the pipeline was higher than the speed of the heavy phase. Over time, one can notice the beginning of the process of formation of the slug mode. At the same time, the gas phase, which is accelerated in the upper part of the pipeline, affects the interface between the two phases in such a way that the process formation of a certain «hump» begins, which subsequently forms a slug. This effect arises since the upper layers of the heavy phase, lying closer to the interface, are accelerated, since the ejection effect arises due to the difference in the velocities of the two phases. Further, there was an increase in the volume of the «bump» of the heavy phase and an increase in its height relative to the initial position of the free surface. It ensures gradual filling of the entire cross-section of the pipe with the heavy phase.

**Table 3.** Pressure fields.

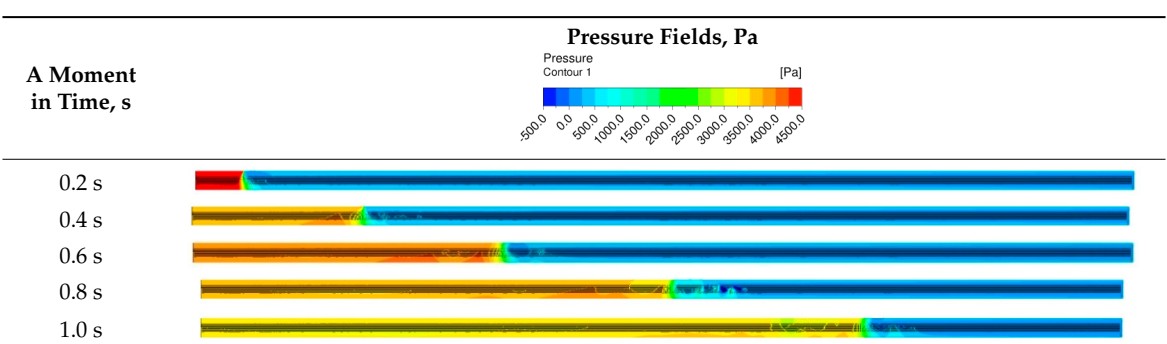

**Table 4.** Phase velocity fields at different points in time.

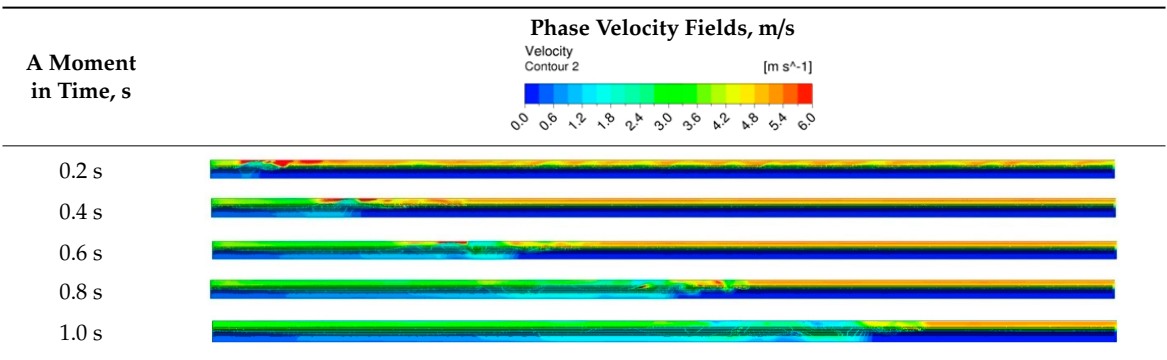

**Table 5.** Volume fraction of the liquid phase in time.

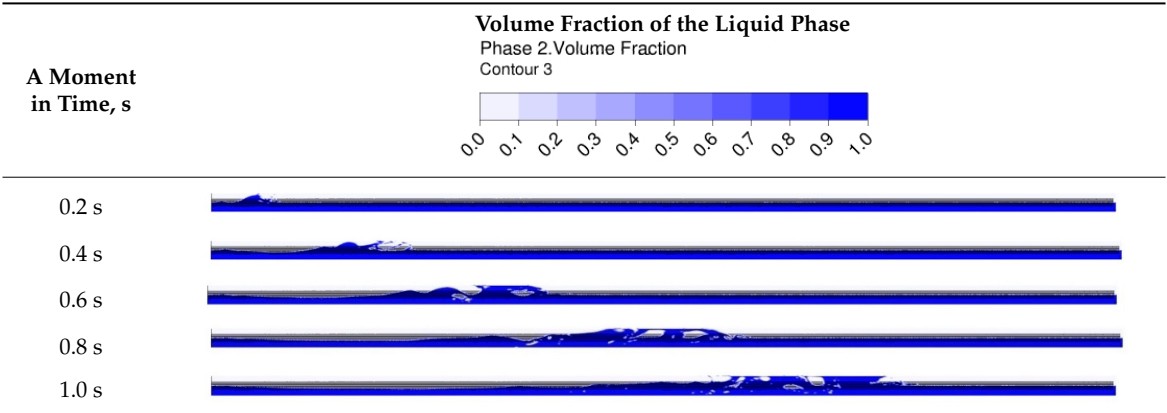

**Table 6.** Pressure fields over time in a system with mesh partitions.

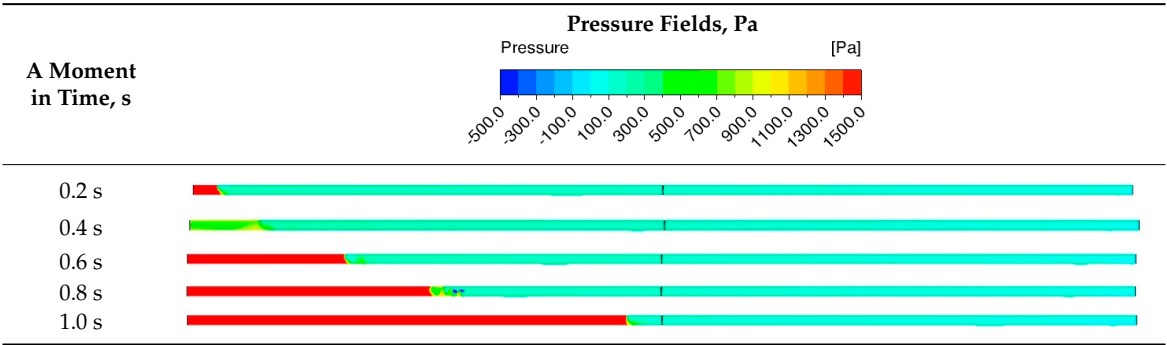

**Table 7.** Phase velocity fields in successive time moments in a system with mesh partitions.

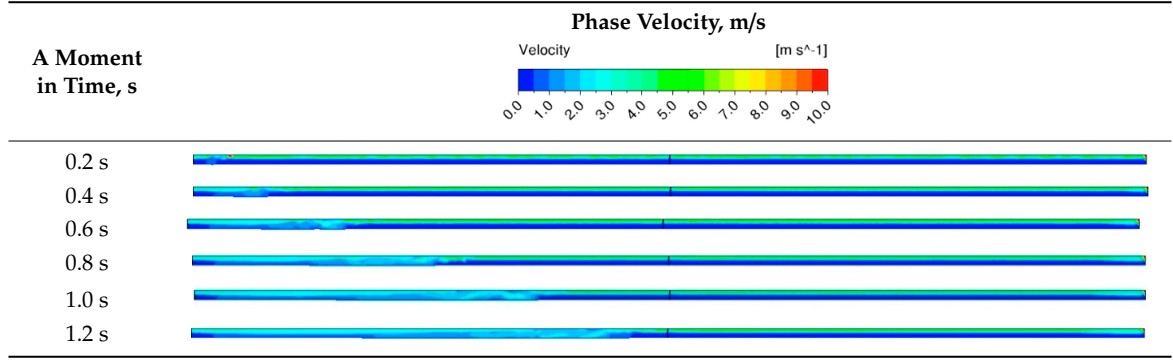

**Table 8.** The volume fraction of the liquid phase in consecutive moments of time in a system with mesh partitions.

| A Moment in Time, s | Volume Fraction of the Liquid Phase |
| --- | --- |
| | Phase 1.Volume Fraction 0.0 0.1 0.2 0.3 0.4 0.5 0.6 0.7 0.8 0.9 1.0 |
| 0.2 s | |
| 0.4 s | |
| 0.6 s | |
| 0.8 s | |
| 1.0 s | |
| 1.2 s | |

One more confirmation of the existence of the slug regime was the trace of three-dimensional streamlines' evolution (Table 5). Streamlines correspond to one of the two subspecies of the slug flow regime, namely the circulation regime.

Without mesh partitions, the slug formation mode began in the middle of the pipe along its length. Further, the slug passed the entire calculated area, which was accompanied by an increase in its volume and its deceleration of motion. This process was accompanied by an increase in pressure upstream, from the left border of the slug. This location of the slug mode beginning explains the choice of the location of the mesh partition's installation.

The evolution of the flow until the slug reached the mesh partition was identical to the flow in an ordinary pipe. However, at the moment of time t = 1.25 s, the collision of the slug with the such promoter was observed. The destruction of the slug regime accompanied the collision, and the flow transit to the layered regime.

## 4. Conclusions

The main objective of the study was to reproduce the slug flow in a circular pipeline using numerical simulation in the ANSYS software package. The process of initiation of the slug flow regime was simulated.

The slug begins to nucleate at a distance of 25% of the pipeline length from the inlet section. At this distance, the slug has poor stability, since its volume is small. In moving downstream, the slug draws in an ever-larger volume of the heavy phase, which makes the regime stable. At a distance of 45–50% from the inlet section, the slug acquires its maximum volume. However, it does not have time to influence the upstream flow characteristics, so there is no significant pressure increase behind the slug. Then, the slug moves downstream to the end of the computational domain. This movement is accompanied by an increase in pressure in the pipeline area before the slug.

The technical solution was proposed to change the flow regime. The solution was the mesh partition that facilitates a change in the flow regime from slug to layered. The location of such promoter was chosen based on the results of calculations performed for the pipeline without it. The mesh partition was installed in the middle of the pipeline. Its resistance was chosen in [39] that pressure drops on such promoter did not exceed 10% of the pressure drop along the length of the pipeline. The proposed technical solution provides a change in the flow regime from slug to layered, which will help to increase the service life and operational safety of real pipelines, at insignificant energy costs, to overcome the additional resistance integrated into the pipeline.

Therefore, further research is needed to address the following challenges:

- Determine the minimum resistance and the shape of the promoter, providing a change in the flow regime.
- Create a method for determining the location of the promoter in the streams.

**Author Contributions:** Conceptualization: V.S., N.V., E.K., D.N., and S.K.; methodology: E.K. and S.K.; software: E.K.; validation: E.K. and D.N.; formal analysis: E.K.; investigation: E.K. and D.N.; resources: E.K.; data curation: S.K.; writing—original draft preparation: N.V.; writing—review and editing: N.V. and V.S.; visualization: E.K.; supervision: V.S.; project administration: D.N.; funding acquisition: V.S. All authors have read and agreed to the published version of the manuscript.

**Funding:** This research work and APC was funded by the Academic Excellence Project 5–100 proposed by Peter the Great St. Petersburg Polytechnic University, St. Petersburg, Russian Federation.

**Conflicts of Interest:** The authors declare no conflict of interest.

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
