# Peer review of "Slug Regime Transitions in a Two-Phase Flow in Horizontal Round Pipe. CFD Simulations"

_applsci, doi:10.3390/app10238739_

Round 1

Reviewer 1 Report

Hello

Please see my comments in the attached file. I only read 5 pages of your paper and with this quality I could not go further and accept it. It seems everything is copied and pasted from another documents without considering the requirements of writing a paper.

Good luck

Author Response

Dear Reviewer,
The authors would like to express their sincere gratitude for the comments that have improved the work. Many comments and recommendations will also be taken into account in future studies.

line 31

power plants, sewage systems and metallurgy

Fixed

line 36

Reference?

Fixed

line 43

Push it to right

Fixed

line 75-76

Where? Is it a literature review?

Fixed

line 75-76

Why didn't you make one single paragraph here?

The paragraphs has combined.

line 90

pathways including 1) .... 2)

Done

line 95

The literature review part was not written very well. Please read other papers and see how they develop this part.

We received several comments on the literature review. We tried to take into account everything.

line 117

this part should be completely edited. It should be write very clear.

The objectives of the study has edited.

line 139-143

Center

Fixed

line 145

where did you explain Figure 4 trough 6?

Fixed

Reviewer 2 Report

In the article entitled: "Transitions of the cochlear regime in a two-phase flow in a horizontal circular tube" the authors Vitaliy Sergeyev, Nikolai Vatin, Evgeny Kotov, Daria Nemova and Svyatoslav Khorobrov presented the results of a numerical simulation of the creation of a slug flow in a horizontal circular tube. Moreover, they proposed a relatively simple and cheap two-phase flow promoter to break up the slugs that are formed. The correctness of operation of such a mesh partition was also checked on the basis of numerical calculations.

The formulation of the model (the set of equations applied) and the methodology of its solution are not objectionable. Therefore, also the correctness of the obtained results does not raise any doubts.

The text is "readable" although I am not a linguist. Nevertheless, some English phrases seem to conflict with applicable rules. So maybe it's worth having a specialist review the text of the article.

A detailed list of shortcomings noticed:

line 19

I suggest replacing “pressure drops” with “pressure drop”.

line 20-21

I suggest replacing “The grid transited the flow regime from slug to layered, which” with “The grid transition of flow regime from slug to layered one”.

line 22

I suggest replacing “cost” with “energy cost”.

line 31

I suggest replacing “[1], [2], [3], [4], [5]” with “[1-5]”.

lines 32-34

The information provided in this paragraph is contradictory. This paragraph should be re-edited.

line 37

 I suggest replacing “gas phase and liquid phases” with “gas and liquid phases”.

line 41

I suggest replacing “modes mainly” with “modes are mainly”.

line 42

I suggest replacing “separated by the channel” with “separated along the channel”.

line 49

Dot should be introduced at the end of the sentence.

line 51

I suggest replacing “Plugged” with “Plug”.

line 53

I suggest replacing “The shells” with “The bubble shells”.

line 57

I suggest replacing “smooth pipes [11]” with “smooth ones [11]”.

line 59

I suggest replacing “slug regime flow-phase gas-liquid flow” with “slug regime of gas-liquid flow”.

line 66

I suggest replacing “streamlines flowing around” with “streamlines drawn around”.

lines 67-68

Consider plotting the direction of the mixture flow in Figure 1. This information may be of interest to less advanced readers. After all, this paragraph is part of the introduction.

line 70

I suggest replacing “1- fluid flow lines in circulation mode; 2 – fluid flow lines in bypass flow mode” with “streamlines of the liquid: 1 – in circulation; 2 – in bypass modes”.

line 72

I suggest replacing “[14], [15], [16]” with “[14-16]”.

line 74

I suggest replacing “[17], [18], [19], [20]” with “[17-20]”.

lines 75-86

For thematic reasons, I suggest combining lines 75-84 into one paragraph.

lines 83-84

I suggest replacing “[22], [23],[24]” with “[22-24]”.

line 92

I suggest replacing “model based” with “model is based”.

line 94

It seems to me that the term “local friction drags” should be replaced by “local drag coefficients”. Please check English terminology.

line 104

I suggest replacing “limited mainly” with “mainly limited”.

line 105

I suggest replacing “[29], [30], [31]” with “[29-31]”.

line 107

The abbreviation VOF (Volume Of Fluid) should be deciphered at this point. However, this is given further on in line 201.

line 119

I suggest replacing "using methods" with “using some of available methods”.

lines 132-133

The text on lines 132-133 is identical to that on lines 129-130. I suggest you delete lines 132-133.

line 135

I suggest replacing "local resistances in" with “local modifications inside”.

line 136

I suggest replacing "grid of plates" with “mesh partition”.

line 137

I suggest replacing “plates” with “partitions”.

line 137

I suggest replacing “the width of the plate is” with “their thickness is”.

lines 137-138

I suggest replacing “, the number of plates is” with “and their number is”.

line 138

I suggest replacing " the plate" with "pipe fragment with mesh partitions".

line 140

I suggest replacing “type 1” with “first version”.

line 142

I suggest replacing “type 2” with “second version”.

line 145

I suggest replacing “Grid of plates” with “Mesh partition construction”.

lines 146-147

I suggest replacing “the grid convergence of the solution” with “solution grid convergence”.

line 151

I suggest replacing “type 1” with “first version”.

line 143

I suggest replacing “type 2” with “second version”.

line 154

I suggest replacing “For Computational grids” with “Every computational grids”.

line 159

I suggest replacing "In the general case" with "In general".

lines 168-170

Viscosity of air and water should be given in SI units.

line 172

I suggest replacing "the media” with "phases".

line 174

I suggest replacing "was used” with "were used". Plural !

line 176

I suggest replacing "both cases” with "both considered cases".

line 178

I suggest replacing "was set” with "was established".

line 179

I suggest replacing "in correspondence to” with "in agreement with".

line 186

I suggest replacing "phase concentration” with "phase concentrations".

line 187

I suggest replacing "Used turbulent flow models” with "Applied turbulent flow models".

line 191

I suggest replacing "Pulse Equation” with "Momentum Equation (according to Navier-Stocks equation)".

lines 190-193

The given equations are commonly known. However, in view of the less advanced reader, the physical meaning of the notations used in them should be given.

lines 196-197

Both lines should be connected by removing the line break 196.

line 196

I suggest replacing “To close RANS used k-ε model with scalable wall functions.” with “To close the RANS calculation scheme, the k-ε model with scalable wall functions was used.”.

line 198

I suggest replacing “molecular viscosity 10” with “molecular viscosity equals to 10”.

line 203

I suggest replacing “the cells” with “the cell”.

line 203

I suggest replacing “the phases” with “the phase”.

line 205

I suggest replacing “was used” with “was improved”.

line 208

II suggest replacing “is unsteady” with “is unsteady in time domain”.

line 209

I suggest replacing “selected” with “adjusted”.

line 220

I suggest replacing “conservatism” with “conservation rules”.

lines 224-225

The physical meaning of the values marked with symbols should be given in the text.

line 227

I suggest replacing “layers” with “intervals”.

line 230

I suggest replacing “Navier-Stokes equations, energy, concentration, free surface” with “Navier-Stokes, energy, concentration, free surface equations”.

line 232

I suggest replacing “Stage” with “Step”.

line 233

There should be a colon (:) at the end of the line.

line 234

I suggest replacing “member” with “term”.

line 235

I suggest replacing “Stage” with “Step”.

line 236

I suggest replacing “satisfied” with “fulfilled”.

line 238

I suggest replacing “Stage” with “Step”.

line 242

I suggest replacing “Stage” with “Step”.

line 243

I suggest replacing “speed” with “velocity”.

line 248

I suggest replacing “Stage” with “Step”.

line 250

I suggest replacing “Stage” with “Step”.

line 251

I suggest replacing “Stage” with “Step”.

line 252

I suggest replacing “the field is used ?n+1” with “the field of ?n+1 is used”.

line 253

I suggest replacing “Stage” with “Step”.

line 254

I suggest replacing “determined” with “determined as”.

line 255

I suggest replacing “refined” with “improved”.

line 257

I suggest replacing “stages” with “steps”.

line 258

I suggest replacing “using the” with “applied”.

line 259

I suggest replacing “using the” with “applied”.

line 259

I suggest replacing “members” with “terms”.

line 262

I suggest replacing “Computing circuit” with “Computational scheme”.

line 264

The acronym PISO should be deciphered in words.

line 264

I suggest replacing “two” with “both”.

line 266

PRESTO! The scheme should be briefly described or a literature reference should be given. The same remark applies to the Geo-reconstruct scheme on line 267.

lines 281-282

I suggest replacing “(without a grid)” with “(without mesh partitions)”.

line 282

I suggest replacing “The construction is done” with “Calculations were performed”.

line 287

Please consider replacement of “the coefficient of hydraulic friction” with “the drag coefficient”.

line 291

I suggest replacing “free-fall acceleration” with “gravity”.

line 294

I suggest replacing “grid Figure 8” with “mesh partition – vide Figure 8”.

line 295

I suggest replacing the inscription in the picture from "presshure" to "pressure". This remark also applies to Figure 9 (line 305).

Line 296

I suggest replacing “pipeline without grid” with “pipeline without mesh partition”.

line 297

I suggest replacing “Pipeline with the grid” with “Pipeline with mesh partition”.

line 298

I suggest replacing "grid" with "mesh partition".

line 298

I suggest replacing “The grid parameters" with "Its parameters".

line 300

Please consider replacement of “the local resistance coefficient” with “ the local drag coefficient”.

line 301

I suggest replacing “grid" with "mesh partition".

line 301

I suggest replacing "and gas is“ with “and gas are described as”.

line 302

I suggest replacing "take as the same as in variant 1 (without grid)" with "is the same as in the previous variant (without mesh partition)".

lines 303-304

I suggest replacing "variant 2 (the pipeline with the grid)" with "considered case (the pipeline with mesh partition)".

line 306

I suggest replacing "(the pipeline with the grid)" with “(the pipeline with the mesh partiton).”.

line 308

I suggest replacing "for two sets" with “for both considered cases”.

line 309

I suggest replacing "a grid" with “mesh partition”.

line 310

I suggest replacing "must be" with “has to be”.

line 316

I suggest replacing "the process of forming a certain «hump» begins" with “the process formation of a certain «hump» begins”.

line 321

I suggest replacing " entire section" with “entire cross-section”.

line 326

I suggest replacing "Pressure fields at different points in time with the installation of the grid" with “Pressure fields over time in a system with mesh partitions”.

line 326

I suggest replacing "Phase velocity fields at different points in time with grating installation" with “Phase velocity fields in successive time moments in a system with mesh partitions”.

line 327

I suggest replacing "Volume fraction of the liquid phase in time with the installation 327 of the grating" with “Volume fraction of the liquid phase in consecutive moments of time in a system with mesh partitions”.

line 333

I suggest replacing "its deceleration" with “its deceleration of motion”.

lines 335-337

This paragraph should be reformulated as it is difficult to understand.

line 353

I suggest replacing "grid" with “mesh partition”.

line 354

I suggest replacing "the grid" with “such promoter”.

line 355

I suggest replacing "a grid" with “it”.

line 355

I suggest replacing "The grid" with “The mesh partition”.

line 356

I suggest replacing "The resistance of the grid" with “Its resistance”.

line 356

I suggest replacing "pressure drops on the grid" with “pressure drops on such promoter”.

line 362

I suggest replacing "the grid" with “the promoter”.

line 364

I suggest replacing "the grid" with “the promoter”.

Author Response

Dear Reviewer,
The authors would like to express their sincere gratitude for the comments that have improved the work. Many comments and recommendations will also be taken into account in future studies.

line 31

power plants, sewage systems and metallurgy

Fixed

line 36

Reference?

Fixed

line 43

Push it to right

Fixed

line 75-76

Where? Is it a literature review?

Fixed

line 75-76

Why didn't you make one single paragraph here?

The paragraphs has combined.

line 90

pathways including 1) .... 2)

Done

line 95

The literature review part was not written very well. Please read other papers and see how they develop this part.

We received several comments on the literature review. We tried to take into account everything.

line 117

this part should be completely edited. It should be write very clear.

The objectives of the study have edited.

line 139-143

Center

Fixed

line 145

where did you explain Figure 4 trough 6?

Fixed

Reviewer 3 Report

Dear Editor, Dear Authors

I am sending my remarks to the reviewed paper. Thank you very much and I am open for discussion about submitted article to improve it as much as possible.

Kind Regards

Author Response

Dear Reviewer,

The authors would like to express their sincere gratitude for the comments that have improved the work. Many comments and recommendations will also be taken into account in future studies.

1) In my opinion the paper should be sent to proofreading - there are several language mistakes. The proofreading would additionally provide the fluency of the whole text. In current situations it sometimes looks like "Slavic English" and different writing styles can be felt (probably due to contributions of many Authors using slightly different styles).

The authors have tried to make the style of presentation consistent and improve the English language. It may need an additional proofreading as a special service of the MDPI publisher.

2) Title of the article can be completed by "CFD simulations".

Added

3) In the abstract I would start from the purpose (well-formulated). In line 18 after RANS, "equations" should be added.

The purpose had added. The word "equations" had added.

4) In the Fig. 1, the letter "b" looks rather like Russian "bukva" «6», which should be corrected.

Fixed

5) Moreover, the same Figure 1 presents different flow patterns for two-phase flows. They rather look like the vertical flow patterns. In case of horizontal flows, the patterns are the following:

Source: J.M. Quiben. Experimental and analytical study of two-phase pressure drops during evaporation in horizontal tubes.

The same patterns were presented in cited [16].

It would be nice to explain the noticeable differences.

The list and visual patterns of horizontal gas-liquid flow regimes are different for different authors. Since the article is aimed only at the projectile mode, the authors consider it possible to get reference [16] without a figure.

6) Line 119 - fragment "pipeline using methods" should be extended by type of methods.

Fixed

7) Lines 129-130 and 132-133 are exactly the same.

Fixed

8) In my opinion, the additional numerical cases should be considered. It would be useful to conduct calculations also for fully unstructured mesh for the case 1 - it would give the information about potential influence of mesh type difference in the grid area.

We are grateful to the reviewer for his advice, which we will take into account in future works.

9) Mesh size sensitivity analysis should be done and presented. Appropriate mesh size sensitivity analysis would significantly reduce computational time.

The authors do not aim to find the mesh size that would ensure mesh independence. The authors used the mesh that satisfy the following criteria. Suppose the refinement of the meshes by half in all three directions does not change the results by more than 1% relative to the previous results. In that case, the mesh is considered acceptable and provides a mesh independent solution.

10) Moreover, I suggest imposing the symmetry of fine-mesh region in the vicinity of the grid plates - i.e. the distance between fine-mesh border and grid plate should be the same on the both sides of the grid.

The approach proposed by the reviewer saves computing resources. We are grateful to the reviewer for his advice, which we will take into account in future works.

11) I recommend to use SI units (see e.g. lines 167-172).

Fixed

12) I suggest changing order in the paragraph 2 - i.e.:

a) geometry and mesh (as it is currently proposed);

b) mathematical model of the problem - it should be extended (e.g. presenting idea of VOF method with appropriate references, its advantages and disadvantages, limitations), symbols should be explained (which is not done currently), description of initial and boundary conditions etc. (values and reason of assumptions), materials description (was the surface tension coefficient applied in Ansys Fluent?). In case of equations - check the language again - it would be better to change the "pulse equation" on "momentum equation"; line 192: instead of "there" - it would be better to use "where". Moreover -wider lexical corrections are needed: e.g. line 234: "non-stationary member" should be changed on "non-stationary term". In the text there is no description of RANS idea and turbulence model description. Authors started quite detailed description of the mathematical model of the problem, but there are some shortcomings. Authors should decide - to describe applied model in details, or simplify the description citing the Fluent theory manual and presenting short information about applied scheme (name), material model and so on. I am open for discussion in this area (just send me an email to discuss this paper: bartosz.fikus@wat.edu.pl).

c) Results and discussion;

d) Conclusions (which are currently wrongly ordered - "1. Conclusions".

We paid much attention to the description of the VOF method, since it is not as well known as the rest of the equations and models used in the article. This is our observation from the articles. The FLUENT description does not provide details on the VOF method.

13) Please, check the applied timestep value (lines 208-210).

Checked. The time step was adaptive. In lines 209-210 its boundaries are written. In fact, it was always close to the minimum value of 10 ms.

14) Please, check the lines: 244-247.

Checked

15) Please, check the numerical schemes presented in p. 2.21.

Checked

16) Why did Authors used analytical approximation of pressure drop along the pipe axis? It is possible to present the numerically obtained pressure distribution. It would be useful to compare it with analytical approximation (Eq. 25). Moreover, Fig. 8 should be corrected -typo in "pressure", axis and their description should be modified.

The comparison of numerically obtained pressure distribution with analytical approximation would not be correct.

We had calculated the pressure and piezometric lines for each phase of the two-phase flow. This calculation aimed to reflect the resistance introduced by the grid visually.

17) How the value of local resistance coefficient for grid was estimated?

We used Handbook of hydraulic resistance.

18) Is the pressure distribution for time t = 0.4 s correct? I think, that the maximum pressure value in this case should be included between the value for 0.2 s and 0.6 s. Am I wrong?

It was an authors’ mistake. We have already fixed it.

19) I recommend to conduct calculations for longer time interval - i.e. up to the moment when the disturbance pass through the pipe length. It would present the direct influence of the grid on the slug. Authors mentioned, that for time t = 1.25 s the interaction between slug and grid is observed, but they did not present results for time greater than 1.2 s.

Authors will definitely follow the recommendation of the reviewer in future research. This kind of computation will take a lot of time and resources. At the same time, the calculations up to the time of 1.25 convincingly showed the effectiveness of our proposed technical solution for the transition the flow regime from slug to bubbly two-phase flow.

Round 2

Reviewer 1 Report

Thanks for updating the contents.

Author Response

Dear Reviewer,
The authors would like to express their sincere gratitude for the manuscript's evaluation and comments that have improved the work. Many comments and recommendations will also be taken into account in future studies.

Reviewer 3 Report

Dear Authors,

Now the paper looks much better. Unfortunately I strongly recommend sending the paper to the proofreading service - it will improve the fluency and language consistency of the article - but I have to underline, that the language was seriously improved.

I still have some doubts and questions which should be explained and improved:

1) In the abstract - Authors used k-ε model / not k-e models;

2) The information about applied mesh size estimation (which was sent in the reply) should be inserted in text;

3) Why "The comparison of numerically obtained pressure distribution with analytical approximation would not be correct."? If you applied the outlet boundary condition - it should be ok. Moreover insead of "input" and "output" you should use "inlet" and "outlet" - that is why the paper needs proofreading;

4) If you used "Handbook of hydraulic resistance" - you should cite it in appropriate place;

5) Timestep mentioned in text "was varied from 10 ms to 0.01 s" - it is exactly the same...

6) Please, write the conservation equations in the RANS form and fulfill it with the turbulence model equations (with constants) and concentration equation - applied in Fluent VOF method and other equations are clearly described in Ansys Fluent Theory Guide. Moreover - you definitely should explain applied symbols (denotations);

7) Please, cite the source of differential schemes presented in paragraph 2.2.

8) Please, correct the Fig. 8 - use e.g. free "Veusz software" to ensure the graph quality. In this form it is not acceptable;

9) Symbol n denotes timestep number and normal vector - it should be corrected - now there is no consistency;

10) Please, check the dynamic viscosity coefficients. E.g. there is probably one order of magnitude error in case of water;

11) In case of Russian word "решетка" - the "grating" or "grid" seem to be a good word (but I know that in case of numerical schemes "grid" has specific meaning).

Thank you and hope for further paper quality improvement.

Kind regards

Author Response

Dear Reviewer,

The authors would like to express their sincere gratitude for the additional comments that have improved the work. Many comments and recommendations will also be taken into account in future studies.

Now the paper looks much better. Unfortunately, I strongly recommend sending the paper to the proofreading service - it will improve the fluency and language consistency of the article - but I have to underline, that the language was seriously improved.

The authors have done an additional language correction.

1) In the abstract - Authors used k-ε model / not k-e models;

Fixed

2) The information about applied mesh size estimation (which was sent in the reply) should be inserted in text;

This information added at lines 172-176.

3) Why «The comparison of numerically obtained pressure distribution with analytical approximation would not be correct.»? If you applied the outlet boundary condition - it should be ok.

The problem is that the article calculates the pressure and piezometric lines for each of the phases separately.

The pressure and piezometric lines are calculated in order to clearly show the smallness of the hydraulic resistance of the mesh partition in comparison with the resistance of the pipeline (Figure 9). 

It is not correct to compare the pressure line or piezometric line of the liquid or gas phase with the results of the analytical approximation of the two-phase flow, because analytical approximation takes into account both phases simultaneously.

Comparison CFD modelling with the analytical approximation of a two-phase flow requires averaging the results of CFD modelling over the volume, and averaging results of CFD modelling over time to obtain a stationary solution. This comparison is another research challenge beyond the scope of this article.

Moreover instead of «input» and «output» you should use «inlet» and «outlet» - that is why the paper needs proofreading;

Fixed at lines 189-198.

4) If you used «Handbook of hydraulic resistance» - you should cite it in appropriate place;

Added reference to [37] Idelchik, I. Handbook of Hydraulic Resistance, Revised and Augmented (2008) Begell House Redding, CT. 861 p.

5) Timestep mentioned in text «was varied from 10 ms to 0.01 s» - it is exactly the same...

The text had changed at line 224-226.

6) Please, write the conservation equations in the RANS form and fulfill it with the turbulence model equations (with constants) and concentration equation - applied in Fluent VOF method and other equations are clearly described in Ansys Fluent Theory Guide.

Moreover - you definitely should explain applied symbols (denotations);

In order not to overload the article with unnecessary generally known equations, the authors added a reference to the Reynolds equations at line 210 [Wilcox, D.C. Turbulence Modeling for CFD (Third Edition); DCW Industries, Inc., 2006; ISBN 978-1-928729-08-2].

The reference to k-ε model had added at line 212 [Jones, W.P.; Launder, B.E. The prediction of laminarization with a two-equation model of turbulence. Int. J. Heat Mass Transf. 1972, 15, 301–314, doi:10.1016/0017-9310(72)90076-2].

The standard constants were used in the k-ε model that Fluent offers by default. That had added at line 215.

The reference to VOF methods had added to line 221 [Hirt, C.W.; Nichols, B.D. Volume of fluid (VOF) method for the dynamics of free boundaries. J. Comput. Phys. 1981, 39, 201–225, doi:10.1016/0021-9991(81)90145-5.]

7) Please, cite the source of differential schemes presented in paragraph 2.2.

The different schemes based on the source [ANSYS Fluent Theory Guide; ANSYS, Inc.: Southpointe, 2016].

The reference on in has added at line 233.

Based on this source, authors had done a more detailed explanation of fluid motion equations’ integration at a one-time step. Also explained that an iterative scheme for simultaneous solution of the Navier-Stokes, energy, concentration, free surface equations could be based on an implicit splitting algorithm in physical variables could be demonstrated by the eight steps.

The text had modified at lines 240-241, 249-251.

8) Please, correct the Fig. 8 - use e.g. free «Veusz software» to ensure the graph quality. In this form it is not acceptable;

Fig. 8 and Fig.9 had changed.

9) Symbol n denotes timestep number and normal vector - it should be corrected - now there is no consistency;

The symbol n has changed to k in formulas (7) – (19). The denotation has added at line 235.

10) Please, check the dynamic viscosity coefficients. E.g. there is probably one order of magnitude error in case of water;

Fixed in lines 183, 184.

11) In case of Russian word «Ñ€ÐµÑˆÐµÑ‚ка» - the «grating» or «grid» seem to be a good word (but I know that in case of numerical schemes «grid» has specific meaning).

It was so in the first manuscript’s version. Then authors had changed in on “mesh partition” in accordance of another reviewer’s recommendation. It seems to authors that the text is understandable enough in this aspect.